# Functional gradients of the cerebellum

Xavier Guell[1,2]*, Jeremy D Schmahmann[2,3], John DE Gabrieli[1], Satrajit S Ghosh[1,4]*

[1]McGovern Institute for Brain Research, Massachusetts Institute of Technology, Cambridge, United States; [2]Laboratory for Neuroanatomy and Cerebellar Neurobiology, Department of Neurology, Massachusetts General Hospital, Harvard Medical School, Boston, United States; [3]Ataxia Unit, Cognitive Behavioral Neurology Unit, Department of Neurology, Massachusetts General Hospital, Harvard Medical School, Boston, United States; [4]Department of Otolaryngology, Harvard Medical School, Boston, United States

**Abstract** A central principle for understanding the cerebral cortex is that macroscale anatomy reflects a functional hierarchy from primary to transmodal processing. In contrast, the central axis of motor and nonmotor macroscale organization in the cerebellum remains unknown. Here we applied diffusion map embedding to resting-state data from the Human Connectome Project dataset (n = 1003), and show for the first time that cerebellar functional regions follow a gradual organization which progresses from primary (motor) to transmodal (DMN, task-unfocused) regions. A secondary axis extends from task-unfocused to task-focused processing. Further, these two principal gradients revealed novel functional properties of the well-established cerebellar double motor representation (lobules I-VI and VIII), and its relationship with the recently described triple nonmotor representation (lobules VI/Crus I, Crus II/VIIB, IX/X). Functional differences exist not only between the two motor but also between the three nonmotor representations, and second motor representation might share functional similarities with third nonmotor representation.
DOI: https://doi.org/10.7554/eLife.36652.001

**\*For correspondence:**
xaviergp@mit.edu (XG);
satra@mit.edu (SSG)

**Competing interests:** The authors declare that no competing interests exist.

## Introduction

Comprehending the relationship between macroscale structure and function is fundamental to understanding the nervous system. One central principle in the study of the cerebral cortex is that macroscale anatomy reflects a functional hierarchy from primary to transmodal processing (*Mesulam, 1998*, *2008*). For example, higher-level aspects of movement planning and decision making are situated predominantly in the anterior aspects of the frontal lobe close to the primary motor cortex, while spatial attention and spatial awareness processes predominantly engage regions of the posterior parietal lobe that are closer to the primary somatosensory cortex (*Andersen and Cui, 2009*). Similarly, higher-level aspects of auditory processing such as language comprehension (e.g. Wernicke's area) are situated closer to the primary auditory cortex, while higher-level aspects of motor processing such as language production (e.g. Broca's area) are situated closer to the primary motor cortex.

In contrast, and despite its growing importance in basic and clinical neuroscience, the central axis of motor and nonmotor macroscale organization in the cerebellum remains unknown. The cerebellum has extensive connectivity with motor and nonmotor aspects of the extracerebellar structures. In addition to anatomy, evidence from clinical, behavioral and neuroimaging studies indicates that the human cerebellum is engaged not only in motor control but also in cognitive and affective processing (*Schmahmann and Pandya, 1991*; *Schmahmann, 1996*; *Baillieux et al., 2008*; *Stoodley and Schmahmann, 2009*; *Thompson and Steinmetz, 2009*; *Tedesco et al., 2011*; *Stoodley et al., 2012*; *E et al., 2014*; *Koziol et al., 2014*; *Guell et al., 2015*; *Hoche et al., 2016*, *2018*; *Schmahmann and Pandya, 1997a*, *Schmahmann and Pandya, 1997b1997b*; *Middleton and*

*Strick, 1994*; *Schmahmann and Sherman, 1998*; *Levisohn et al., 2000*; *Riva and Giorgi, 2000*; *Ravizza et al., 2006*; *Schmahmann et al., 2007*). Further, structural and functional analyses have identified cerebellar abnormalities not only in primary cerebellar injury or degeneration, but also in many psychiatric and neurological diseases that degrade cognition and affect. Examples include major depressive disorder, anxiety disorders, bipolar disorder, schizophrenia, attention deficit and hyperactivity disorder, autism spectrum disorder (*Phillips et al., 2015*; *Arnold Anteraper et al., 2018*), posttraumatic stress disorder (*Wang et al., 2016*), fibromyalgia (*Kim et al., 2015*), Alzheimer's disease (*Guo et al., 2016*), frontotemporal dementia (*Guo et al., 2016*), vascular dementia (*Bastos Leite et al., 2006*), Huntington's disease (*Wolf et al., 2015*), multiple sclerosis (*Wilkins, 2017*) and Parkinson's disease (*Wu and Hallett, 2013*). Unmasking the basic hierarchical principles of cerebellar macroscale organization can therefore have large impact in basic and clinical neuroscience.

The study of connectivity gradients in resting state fMRI data - an aspect of cerebellar functional neuroanatomy that remains largely unexplored - can provide critical information necessary to address this knowledge gap. The fact that there are no cerebellar cortical association fibers (*Schmahmann, 1996*; *Schmahmann and Pandya, 2008*) makes it difficult to analyze intra-cerebellar progressive hierarchical relationships using anatomical techniques. Resting-state functional connectivity from fMRI data becomes, in this case, a useful approach to interrogate functional relationships between nearby cerebellar structures which are not directly connected. Contrasting with the common practice of partitioning neural structures into discrete areas with sharp boundaries (*Damoiseaux et al., 2006*; *Yeo et al., 2011*), Margulies and colleagues (*Margulies et al., 2016*) provided a simple and powerful description of the 'principal gradient' of resting-state functional connectivity in the cerebral cortex using diffusion map embedding. This gradient extended from primary/unimodal cortices to regions corresponding to the default mode network (DMN), confirming the primary-unimodal-transmodal hierarchical principle of the cerebral cortex (*Mesulam, 1998*, *2008*). Similarly, Sepulcre and colleagues (*Sepulcre et al., 2012*) revealed transitions from primary sensory cortices to higher-order brain systems using stepwise functional connectivity. The present study is the first to use these analyses in the cerebellum.

Here we set out to describe the principal gradients of intra-cerebellar connectivity by using resting-state diffusion map embedding. We aim to unmask the central axis of motor and nonmotor macroscale organization of the cerebellum, analogous to the fundamental primary-unimodal-transmodal hierarchical principle of cerebral cortex (*Mesulam, 1998*, *2008*). To further characterize the functional significance and implications of these continuous gradients, we aimed to analyze their relationship with discrete cerebellar parcellations including task activity maps, resting state maps, and distinct areas of motor (first = I-VI, second = VIII) and nonmotor representation (first = VI/Crus I, second = Crus II/VIIB, third = IX/X) (*Buckner et al., 2011*; *Guell et al., 2018a*). We took advantage of the newly available and unparalleled power of the Human Connectome Project (HCP) dataset, where each participant (n = 1003) provided one full hour of resting-state data. We incorporated task activity maps (motor, working memory, emotion, social, and language processing) from a previously analyzed subset of the same group of participants (*Guell et al., 2018a*) (n = 787). Maps of cerebellar representation of cerebral cortical resting-state networks were obtained from the study of *Buckner et al. (2011)*, calculated in a different group of participants (n = 1000). Data-driven clustering and stability analyses were used to compare our findings with previous discrete cerebellar parcellations, as well as to validate our hypothesis-driven divisions. A supplementary analysis of cerebello-cerebral connectivity was used to validate our interpretation of asymmetries between the two motor and three nonmotor regions of cerebellar representation. Analysis of single participants from our cohort tested the robustness of our findings at the individual level, and a parallel analysis of functional gradients based on connectivity from the cerebellum to the cerebral cortex investigated the relationship between intra-cerebellar and cerebello-cerebral principles of organization.

## Results

Our analyses included data from 1003 participants of the Human Connectome Project (HCP) (*Van Essen et al., 2013*). We calculated functional gradients by analyzing the similarity of intra-cerebellar resting-state functional connectivity patterns of all cerebellar data points using diffusion map embedding (*Figure 1—figure supplement 1*). The principal component resulting from this analysis

(gradient 1) captures the main axis of macroscale functional organization of the cerebellum, and additional orthogonal components (gradient 2, 3, etc.) capture additional functional organizational properties. The resulting gradients were interpreted by analyzing their relationship to task activity (*Guell et al., 2018a*) and resting-state network (*Buckner et al., 2011*) cerebellar maps from previous studies, explored and compared to previous cerebellar parcellations using clustering analyses, and also compared to functional gradients calculated in the cerebellum using connectivity data between the cerebellum and the cerebral cortex (rather than intra-cerebellar connectivity data). We also analyzed asymmetries between the two motor (I-VI, VIII) and three nonmotor regions of cerebellar representation (VI/Crus I, Crus II/VIIB, IX/X) (*Guell et al., 2018a*) by comparing their relative position along gradients 1 and 2. As a supplementary analysis, we also contrasted cerebello-cerebral connectivity from each of these areas of representation. Analyses using single-subject data rather than group-averaged data tested whether our findings remained observable at the individual subject level.

## Cerebellum gradients and relationship with discrete task activity and resting-state maps

Gradient 1 explained the largest part of variability in resting-state connectivity patterns within the cerebellum (*Figure 1A*). It extended bilaterally from lobules IV/V/VI and lobule VIII to posterior aspects of Crus I and Crus II as well as medial regions of lobule IX. Overlap with task activity maps (*Figure 1B*) revealed that Gradient 1 is anchored at one end by cerebellar motor regions and at the other end by regions engaged in the language task of the HCP dataset. Regions situated between the two extreme ends corresponded to areas involved in working memory and emotion task processing. Social processing was diffusely distributed across Gradient 1. Overlap with cerebellar representations of cerebral cortical resting-state networks (*Figure 1B*) revealed that Gradient 1 extends from sensorimotor network to DMN regions of the cerebellum. Ventral/dorsal attention and fronto-parietal networks were situated between the two extreme ends.

Gradient 2, the component accounting for the second-most variance, included at one end the anterior portions of Crus I and Crus II bilaterally (*Figure 1A*). These regions corresponded to areas engaged in the HCP working-memory task (*Figure 1B*). The same areas were included in cerebellar representations of the frontoparietal resting-state network. The other end of Gradient 2 included both regions involved in motor processing and regions involved in language processing; these areas correspond, respectively, to sensorimotor network and DMN regions. Additional gradients are shown in *Figure 1—figure supplement 2*.

Data from a single participant revealed a similar distribution of gradients 1 and 2 and a similar relationship with the same single subject motor, language and working-memory task processing (*Figure 1—figure supplement 3*). Functional gradients calculated using concatenated and normalized time series of 32 unrelated participants also revealed a similar distribution of gradients 1 and 2 (*Figure 1—figure supplement 4*). Within this group, individual subjects revealed gradients with a similar distribution in most cases for gradient 1 (29 out of 32 participants), and in half of the cases for gradient 2. Future studies aiming to perform group comparison statistics using functional gradients might benefit from alternate alignment strategies (see details in legend of *Figure 1—figure supplement 4*).

Functional gradients calculated using functional connectivity values from the cerebellum to the cerebral cortex (rather than from the cerebellum to the cerebellum) revealed a remarkably similar distribution (*Figure 2*). In addition, clustering of connectivity gradients revealed discrete networks similar to cerebello-cerebral connectivity parcellations from *Buckner et al. (2011)* (*Figure 1—figure supplement 5*).

## Investigation of individual areas of motor and nonmotor representation

Resting-state as well as task processing analyses have revealed a cerebellar double motor (lobules I-VI and VIII) and triple non-motor representation (lobules VI/Crus I, Crus II/VIIB and IX/X) (*Buckner et al., 2011*; *Guell et al., 2018a*), but the functional significance of this distribution remains unknown. To investigate individual areas of motor and nonmotor representation, we isolated Gradient 1 highest 5% voxels within each area of nonmotor representation ('*High-G1*'), Gradient 2 highest 5% voxels within each area of nonmotor representation ('*High-G2*'), and Gradient 1 lowest 5% voxels

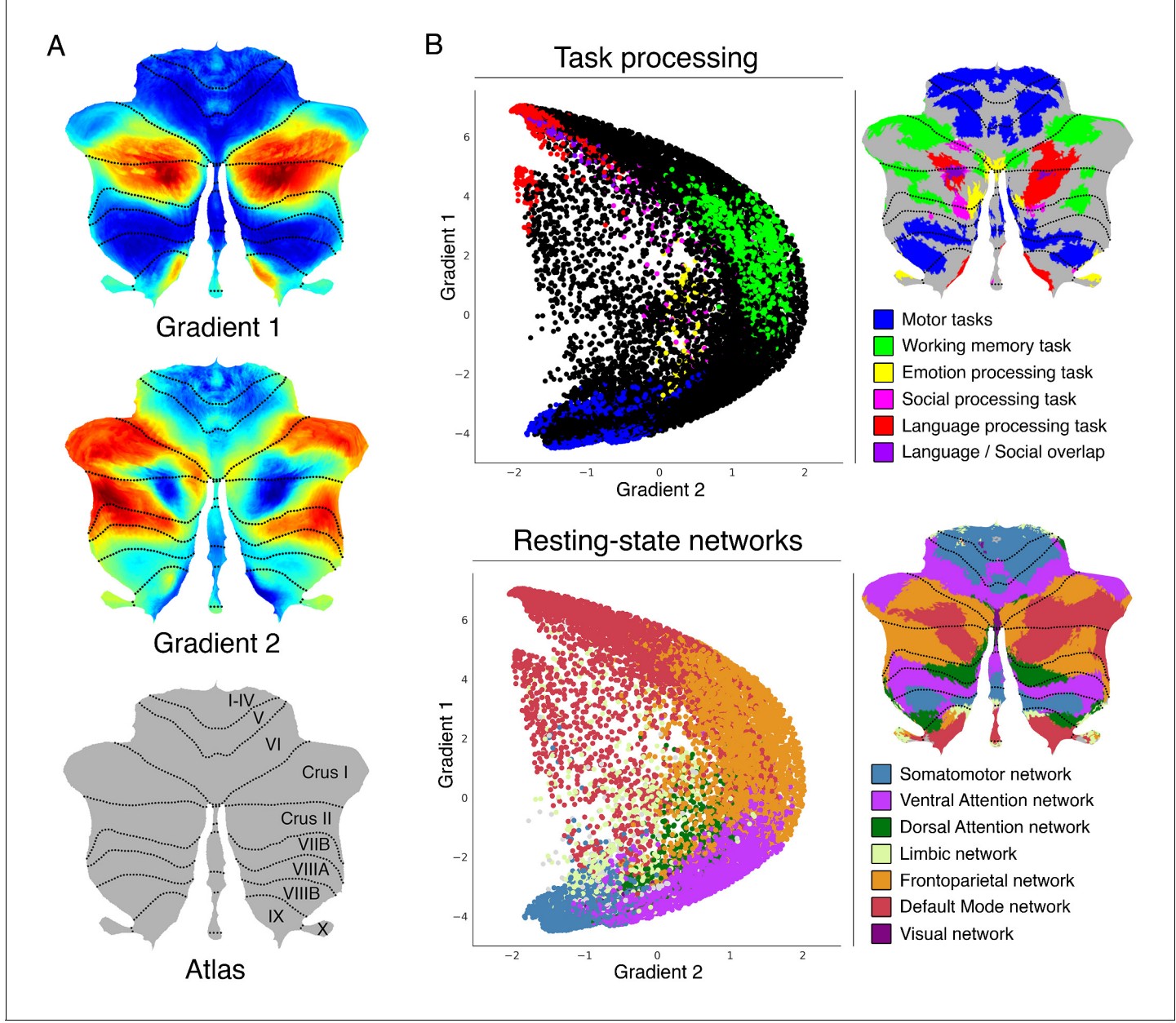

**Figure 1.** Cerebellum gradients and relationship with discrete task activity maps (from *Guell et al., 2018a*) and resting-state maps (from *Buckner et al., 2011*). Gradient 1 extended from language task/DMN to motor regions. Gradient 2 isolated working memory/frontoparietal network areas. (A) Cerebellum flatmap atlas and gradients 1 and 2. (B) A scatterplot of the first two gradients. Each dot corresponds to a cerebellar voxel, position of each dot along x and y axis corresponds to position along Gradient 1 and Gradient 2 for that cerebellar voxel, and color of the dot corresponds to task activity (top) or resting-state network (bottom) associated with that particular voxel.

DOI: https://doi.org/10.7554/eLife.36652.002

The following figure supplements are available for figure 1:

**Figure supplement 1.** Schematic representation of diffusion map embedding.
DOI: https://doi.org/10.7554/eLife.36652.003
**Figure supplement 2.** Additional gradients.
DOI: https://doi.org/10.7554/eLife.36652.004
**Figure supplement 3.** Cerebellum gradients and relationship with discrete task activity for an individual subject (one resting-state run of 15 min).
DOI: https://doi.org/10.7554/eLife.36652.005
**Figure supplement 4.** Cerebellum single-subject and group-level functional gradients calculations in a group of 32 participants.
DOI: https://doi.org/10.7554/eLife.36652.006

*Figure 1 continued on next page*

*Figure 1 continued*

**Figure supplement 5.** Clustering of connectivity gradients revealed discrete networks similar to cerebello-cerebral connectivity parcellations from *Buckner et al. (2011)*.

DOI: https://doi.org/10.7554/eLife.36652.007

**Figure supplement 6.** Cerebral cortical resting-state networks from Yeo and colleagues (*Yeo et al., 2011*) (dark purple, visual; blue, somatomotor; green, dorsal attention; violet, ventral attention; cream, limbic; orange, frontoparietal; red, default network) revealed an overlap between DMN (red) and language task activity (grey) also in the cerebral cortex.

DOI: https://doi.org/10.7554/eLife.36652.008

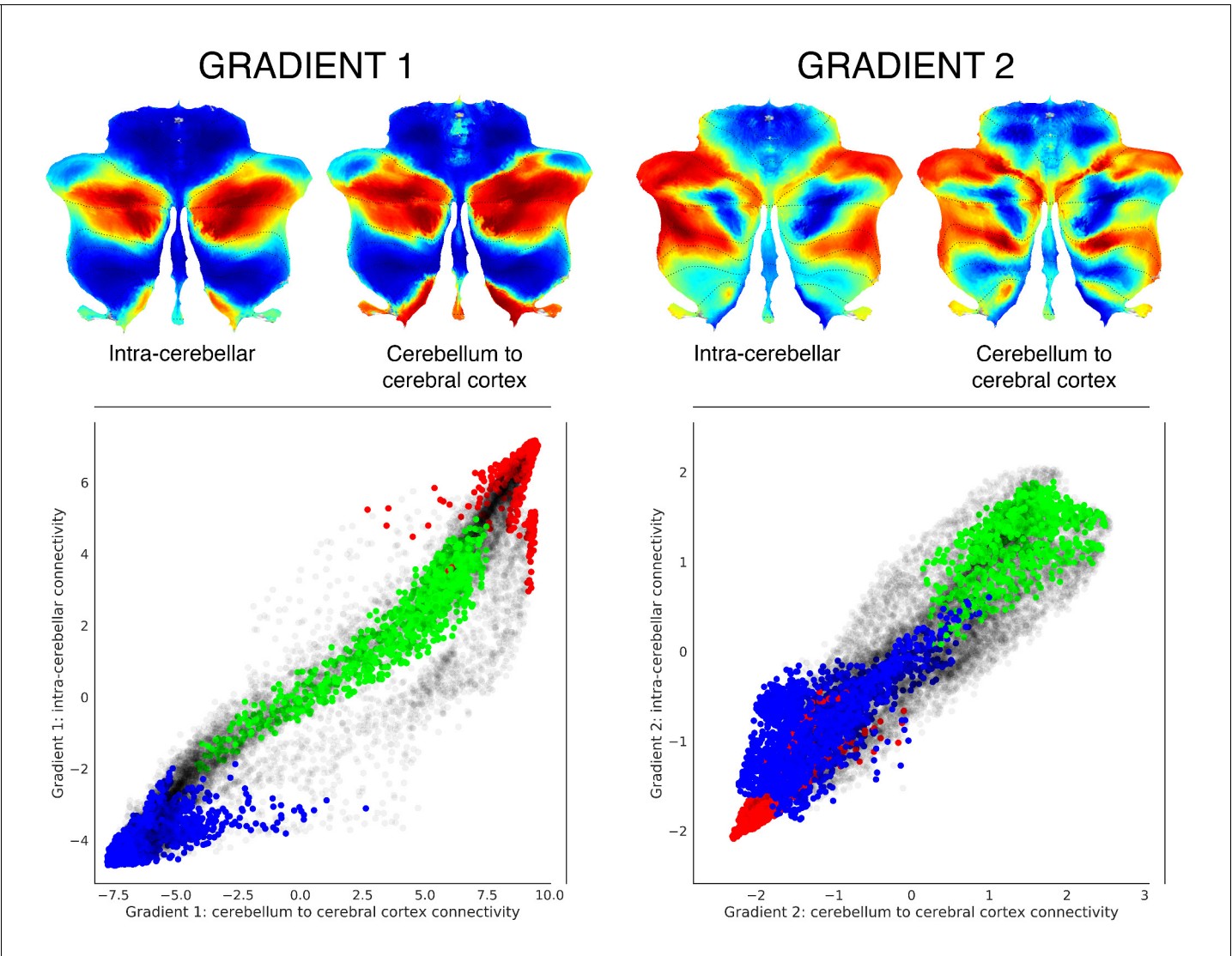

**Figure 2.** Functional gradients calculated based on functional connectivity between the cerebellum and the cerebral cortex revealed a similar distribution when compared to intra-cerebellar functional gradients. Left scatterplot represents intra-cerebellar gradient 1 (y axis) vs. cerebellum-to-cerebral-cortex gradient 1 (x axis). Right scatterplot represents intra-cerebellar gradient 2 (y axis) vs. cerebellum-to-cerebral-cortex gradient 2 (x axis). Scatterplot colors correspond to cerebellar task activation maps as shown in *Figure 1*: motor (blue), working memory (task-focused cognitive processing) (green), and language (task-unfocused cognitive processing) (red). As in the case of intra-cerebellar functional gradient 1 (left y axis), cerebello-cortical gradient 1 (left x axis) distinguishes motor (blue) vs. task-positive cognitive processing (green) vs. task-negative cognitive processing (red). Similarly, as in the case of intra-cerebellar functional gradient 2 (right y axis), cerebello-cortical gradient 2 (right x axis) isolates task-positive cognitive processing (green).

DOI: https://doi.org/10.7554/eLife.36652.009

within each area of motor representation ('*Low-G1*') (*Figure 3A*). This parcellation is functionally meaningful because High-G1/High-G2 correspond to different nonmotor task activity and resting-

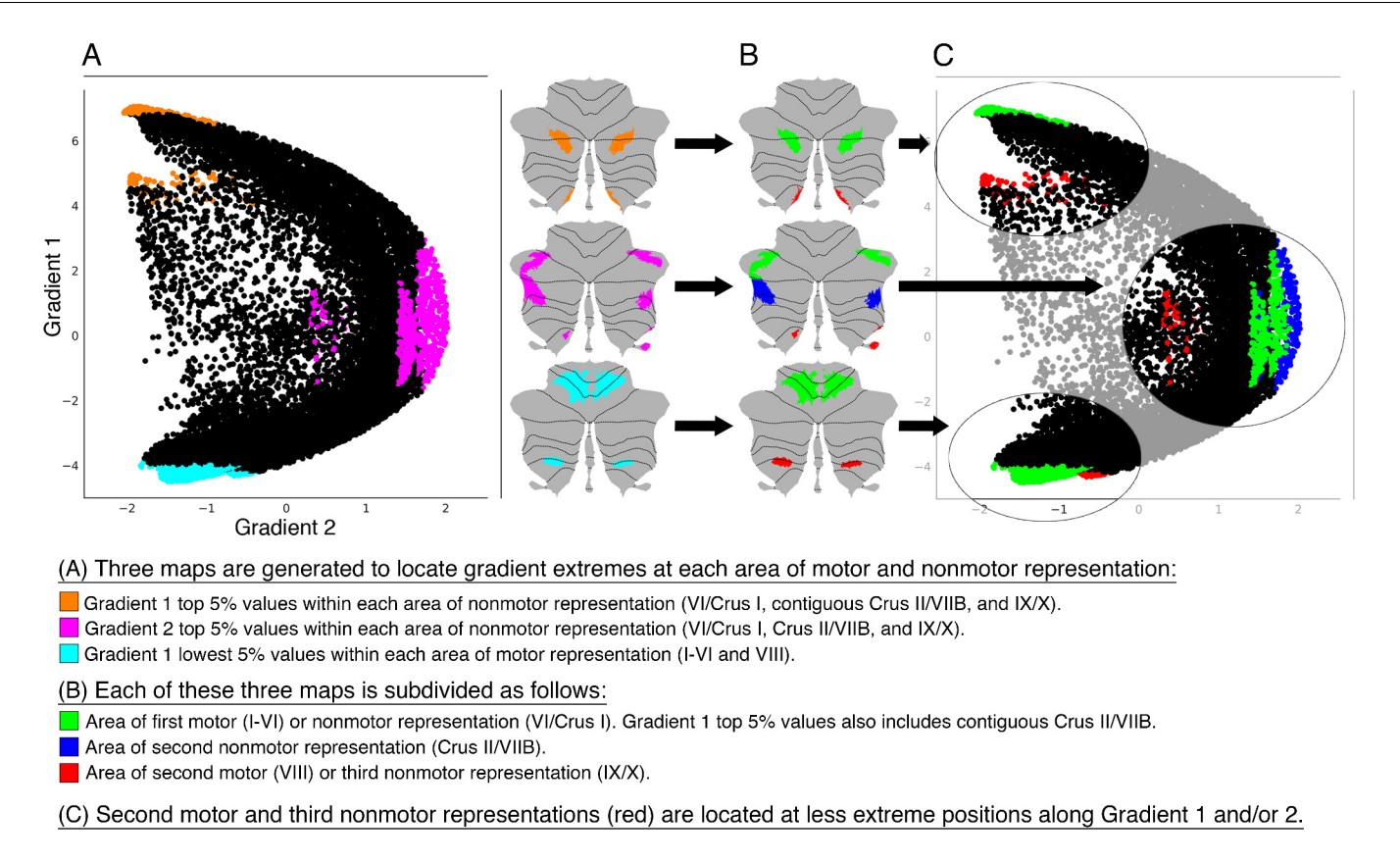

(A) Three maps are generated to locate gradient extremes at each area of motor and nonmotor representation:

■ Gradient 1 top 5% values within each area of nonmotor representation (VI/Crus I, contiguous Crus II/VIIB, and IX/X).
■ Gradient 2 top 5% values within each area of nonmotor representation (VI/Crus I, Crus II/VIIB, and IX/X).
■ Gradient 1 lowest 5% values within each area of motor representation (I-VI and VIII).

(B) Each of these three maps is subdivided as follows:

■ Area of first motor (I-VI) or nonmotor representation (VI/Crus I). Gradient 1 top 5% values also includes contiguous Crus II/VIIB.
■ Area of second nonmotor representation (Crus II/VIIB).
■ Area of second motor (VIII) or third nonmotor representation (IX/X).

(C) Second motor and third nonmotor representations (red) are located at less extreme positions along Gradient 1 and/or 2.

**Figure 3.** Investigation of individual areas of motor and nonmotor representation. Second motor and third nonmotor representations (shown in red) were consistently located at less extreme positions along Gradient 1 and/or 2. This observation suggests that second motor representation is functionally distinct from first motor representation, and that third nonmotor representation is functionally distinct from first/second nonmotor representation. Because second motor and third nonmotor representations (shown in red) were consistently located at less extreme positions along Gradient 1 and/or 2, differences between second and first motor representation might be similar to the differences between third and first/second nonmotor representations. Specifically, second motor and third nonmotor representations might both correspond to a less extreme level of information processing (namely, information processing at a functional space which is distant from Gradient 1 or Gradient 2 extreme values). The same relationship was observed in task and resting-state network maps (*Figure 3—figure supplement 1*), as well as when using thresholds other than 5% (from 0.5% to 15%, *Figure 3—figure supplement 2*).

DOI: https://doi.org/10.7554/eLife.36652.010

The following figure supplements are available for figure 3:

**Figure supplement 1.** Distribution shown in *Figure 3C* can also be observed when using task activity maps (from *Guell et al., 2018a*) or resting-state network maps (from *Buckner et al., 2011*).

DOI: https://doi.org/10.7554/eLife.36652.011

**Figure supplement 2.** Distribution shown in *Figure 3C* can also be observed when using thresholds other than 5%.

DOI: https://doi.org/10.7554/eLife.36652.012

**Figure supplement 3.** Clustering analyses validate our hypothesis-driven High-G1/High-G2/Low-G1 division.

DOI: https://doi.org/10.7554/eLife.36652.013

**Figure supplement 4.** Investigation of individual areas of motor and nonmotor representation using data from one single subject (one resting-state run of 15 min).

DOI: https://doi.org/10.7554/eLife.36652.014

**Figure supplement 5.** Contrasts of cerebello-cerebral connectivity from Gradient 1 or 2 peaks at each area of motor or nonmotor representation.

DOI: https://doi.org/10.7554/eLife.36652.015

state network maps (language and DMN vs. working memory and frontoparietal/ventral-dorsal attention), and Low-G1 corresponds to areas of motor processing (*Figure 1*).

Further, we isolated each individual representation in task activity maps from *Guell et al. (2018a)* and resting-state network maps from *Buckner et al. (2011)*. Specifically, we isolated first motor (I-VI) and second motor representation (VIII) of the motor task map and somatomotor network. Language task and DMN were separated in first and contiguous second nonmotor (VI/Crus I/Crus II/VIIB) and third nonmotor (IX/X) representations, given the contiguous first and second representations of these maps in Crus I/Crus II. All other tasks and resting-state maps were divided in first nonmotor (VI/Crus I), second nonmotor (Crus II/VIIB) and third nonmotor representation (IX/X).

When analyzing the position of each individual representation along Gradient 1 and 2, second motor and third nonmotor representations were consistently located at less extreme positions along these two gradients. We observed this consistently across gradient-derived parcellations (High-G1/Hig-G2/Low-G1, *Figure 3*) as well as task and resting-state network maps (*Figure 3—figure supplement 1*). Second motor and third nonmotor representations are shown in red in *Figure 3* and *Figure 3—figure supplement 1*. Note, for example, that second motor representation in Low-G1, motor task, and somatomotor network was located at a less extreme position along Gradient 2. Following a similar logic, third representation of High-G1, language task, and DMN was located at a less extreme position along Gradient 1. Third representation of High-G2 and frontoparietal network was located at a less extreme position along Gradient 2. Third representation of emotion, social task, ventral attention, dorsal attention, and limbic networks showed a less clear distribution, but was nonetheless consistently located at more central (i.e. less extreme) position along Gradient 1 and/or 2. This organization could not be observed in working memory task and visual network given that these maps were not represented in lobules IX/X.

A data-driven clustering of the first two gradients resulted in a division of gradients 1 and 2 in three areas encompassing our High-G1/High-G2/Low-G1 parcellation (*Figure 3—figure supplement 3*), further supporting this hypothesis-driven division. The same relationship between the two motor and three nonmotor areas of representation was observed in the analysis of a single subject with only one resting-state run of 15 min (*Figure 3—figure supplement 4*). A supplementary cerebello-cerebral connectivity analysis revealed additional differences in cerebral cortical connectivity from each area of representation (*Figure 3—figure supplement 5*).

## Discussion

This is the first study to investigate the progressive, hierarchical organization of the cerebellum. Contrasting with the fundamental and well-established primary-unimodal-transmodal hierarchical organization in the cerebral cortex (*Mesulam, 1998*, *2008*), the principal axis of cerebellar motor and nonmotor organization remains unknown. We describe for the first time that cerebellar functional regions follow a gradual organization which progresses from primary (motor) to transmodal (DMN, task-unfocused) regions. Further, the relationship between the two principal gradients and the two motor and three nonmotor areas of representation revealed for the first time that there are functional differences not only between the two motor but also between the three nonmotor areas of representation. An initial novel hypothesis regarding the nature of these differences is generated by noting that nonmotor processing in lobules IX/X (third nonmotor representation) might share functional similarities with motor processing in lobule VIII (second motor representation). These interpretations are further supported by data-driven clustering and cerebello-cerebral functional connectivity analyses. These findings, from an exceptionally large and high-quality dataset, provide new and fundamental insights into the functional organization of the human cerebellum, unmask new testable questions for future studies, and yield an unprecedented tool for the topographical interpretation of cerebellar findings.

### Gradient 1 extends from motor to nonmotor areas: cerebellar macroscale organization is sensorimotor-fugal

Gradient 1 extended from regions corresponding to motor task activity and sensorimotor network representation to regions corresponding to language task activity and DMN representation (*Figure 1B*). The overlap between language task activity and DMN may be due to the language task contrast which subtracted listening to stories minus answering arithmetic questions. This subtraction

may capture processes similar to those that engage DMN regions, such as autobiographical memory retrieval, daydreaming, and conceiving the perspective of others (*Buckner et al., 2008*). Consistent with this hypothesis, HCP language task activity also overlapped with DMN in the cerebral cortex (*Figure 1—figure supplement 6*). Working memory task processing was situated at a middle point along Gradient 1, similar to the distribution of frontoparietal and ventral attention networks (*Figure 1B*). It is reasonable to conceptualize working memory as a nonmotor task which is not as distant from motor function as a story listening task, justifying its middle position along Gradient 1. Similarly, tasks that activate DMN regions such as daydreaming and mind wandering (*Mason et al., 2007*; *Christoff et al., 2009*; *Stawarczyk et al., 2011*) can be conceptualized as more distant from motor processing than goal-directed cognitive control and decision-making processes that activate frontoparietal network regions (*Vincent et al., 2008*). Ventral and dorsal attention networks were located far from DMN along Gradient 1 (*Figure 1B*), consistent with the view that DMN and ventral/dorsal attention networks are two opposing brain systems (*Fox et al., 2005*). The frontoparietal network is conceptualized as a mediator between the two (*Vincent et al., 2008*), justifying its position between ventral/dorsal attention networks and DMN along Gradient 1 (*Figure 1B*). This conceptualization of Gradient 1 is also coherent with a previous report analyzing cerebellar activity at multiple time points, from motor planning to motor output (*Hülsmann et al., 2003*). The authors described a lateromedial succession '*from will to action*' which is in accordance with the direction of Gradient 1 from nonmotor to motor regions in our analysis.

This is the first study to report a sensorimotor-fugal macroscale organization in the cerebellum; that is, a hierarchical organization that progresses away from sensorimotor function. A different study using diffusion map embedding analysis in the cerebral cortex reported similar results (*Margulies et al., 2016*). In that case, the principal gradient extended from primary cortices (visual, somatosensory/motor, and auditory) to regions corresponding to the DMN. As in the cerebellum, the frontoparietal network was also located between DMN and ventral/dorsal attention networks, and working memory task activity was also located at a middle position along the principal gradient. Of note, the cerebellum does not show functional connectivity with primary visual or auditory cortices (*Buckner et al., 2011*), but is anatomically and functionally connected with areas of primary sensorimotor processing and consistently engaged in simple motor tasks. It is therefore reasonable to consider that a gradient from motor to DMN areas in the cerebellum is the equivalent of a gradient from motor/visual/auditory to DMN areas in the cerebral cortex.

This finding strongly suggests that cerebellum and cerebral cortex share a similar macroscale principle of organization, namely, that both structures share a hierarchical organization which gradually progresses away from unimodal streams of information processing. While this organization has long been defended in the cerebral cortex (*Mesulam, 1998*; *Margulies et al., 2016*; *Sepulcre et al., 2012*), the present analysis is the first to reveal an analogous principle in the cerebellum. This is a notable observation for two reasons. First, gradients obtained in our analysis correspond to intrinsic connectivity profiles of cerebellar voxels with the rest of the cerebellum only, rather than with the rest of the brain. Therefore, our analysis reflects the organization of the cerebellum without invoking its connectivity profiles with the cerebral hemispheres. In this way, the fact that we observed a similar principle of organization between the cerebral cortex and the cerebellum does not constitute an imposition of cerebro-cerebellar connectivity on our method of analysis (unlike in *Buckner et al., 2011*). Second, the notion that there is a hierarchical organization in the cerebral cortex which gradually progresses away from unimodal streams of information (*Mesulam, 1998*; *Margulies et al., 2016*; *Sepulcre et al., 2012*) is implicitly predicated on the anatomical knowledge that there are synapses linking adjacent cerebral cortical regions (*Schmahmann and Pandya, 2009*). Unlike the cerebral cortex, there are no short or long cerebellar cortical association fibers linking adjacent or distant cerebellar cortical areas with each other (*Schmahmann, 1996*; *Schmahmann and Pandya, 2008*). It is therefore nontrivial to observe this parallel organization in the cerebellum. Our hypothesis is that such a functional organization is a consequence of the arrangement of cerebello-cerebral anatomical connections which in turn affect correlations in resting-state activity between cerebellar regions. Consistent with this possibility, cerebellar functional gradients calculated from cerebello-cerebral connectivity revealed a distribution similar to the intra-cerebellar functional gradients (*Figure 2*). The same possibility raises further questions regarding the precise distribution of cerebello-cerebral anatomical connections that would be required to achieve such a parallel mapping of functional gradients in the cerebral cortex and cerebellum.

The finding that a similar distribution of the first two gradients and their relationship with motor, language and working memory task processing can be observed at an individual level (*Figure 1—figure supplement 3*) supports the assertion that this organization is not an artifact generated as a result of averaging a large number of subjects, and highlights the potential application of this fundamental principle in future small group or single subject investigations (see *Figure 1—figure supplement 4*).

## Integrating gradient 1 and gradient 2: task processing in the cerebellum understood in terms of distance from motor processing and amount of task-focus

Gradient 1 extended from motor to nonmotor (task-unfocused, DMN) regions. In contrast, Gradient 2 (the component accounting for the second-most variance) was anchored at one end by working memory task and frontoparietal network regions. The other extreme of Gradient 2 corresponded to both extremes of Gradient 1, namely, (i) regions corresponding to motor task activity and sensorimotor network and (ii) regions corresponding to language task activity and DMN representation. The functional significance of this distribution might be analyzed as follows. Working memory HCP task corresponds to *Two back* (respond if current stimulus matches the item two back) minus *Zero back* (respond if current stimulus matches target cue presented at start of block). HCP language task correspond to *Story* (listen to stories) minus *Math* (answer arithmetic questions). HCP motor tasks corresponds to *Movement* (tap left fingers, or tap right fingers, or squeeze right toes, or squeeze left toes, or move tongue) minus *Average* (average of the other four movements). What dimension corresponding to the working memory task is equally absent in the language and the motor task contrasts? One possible explanation is task focus. Whereas the working memory task contrast isolates a higher load of working memory (therefore a higher load of task focus), task focus is eliminated from the language task contrast after subtracting the math condition, and task focus is eliminated from the motor task contrast after subtracting the average of other movements. Coherently, frontoparietal and ventral attention networks (the extreme of Gradient 2, *Figure 1B*) are task-positive networks (*Vincent et al., 2008*) while DMN and somatosensory network (the other extreme of Gradient 2) are not.

In this way, Gradient 1 and Gradient 2 classify information processing in the cerebellum along two dimensions: distance from motor processing (Gradient 1) and amount of task-focus (Gradient 2). HCP motor task contrast isolates pure motor processing and eliminates task-focus demands. In consequence, HCP motor task is situated at a minimal position in Gradient 1 (i.e. maximally motor) and at a minimal position in Gradient 2 (i.e. minimally task-focused) (*Figure 1B*). HCP working memory task isolates a higher load of working memory by subtracting a two-back minus a zero-back condition. The isolated cognitive process is closely related to task focus and is therefore situated at a maximum position in Gradient 2. At the same time, working memory represents a nonmotor process and is therefore situated higher than the HCP motor task along Gradient 1. This notwithstanding, working memory is situated lower than the HCP language task along Gradient 1. This order seems logical by considering that goal-nondirected processes targeted by the HCP language task contrast are more distant from pure motor processing than those goal-directed processes isolated by the working memory task contrast. Similarly, mind-wandering states are, by definition, task-unfocused, explaining the position of the HCP language task at the lowest extreme of Gradient 2.

Our interpretation of task focus in the cerebellum in terms of distance from motor processing and amount of task-focus is also coherent with the general distribution of data points when plotting Gradient 1 against Gradient 2 (see plots in *Figure 1B*). First, there are no cerebellar voxels with simultaneous maximum Gradient 1 and Gradient 2 values. Maximum Gradient 1 values correspond to DMN regions, and DMN processes are task-unfocused by definition. Therefore, Gradient 1 maximum values must have low Gradient 2 values. Second, there are no cerebellar voxels with simultaneous minimum Gradient 1 and maximum Gradient 2 values. This distribution is consistent with the notion that increasing attentional demands of a motor task adds nonmotor computational demands. Accordingly, Gradient 1 lowest values cannot increase their position along Gradient 2 without simultaneously acquiring a higher position along Gradient 1.

While HCP motor, working memory, and language task activity maps were situated at extreme regions along Gradient 1 and/or 2 (*Figure 1B*), social and emotion processing did not adhere to any extreme along these gradients. One possibility is that functional gradients 1 and 2 fail to capture

relevant aspects pertaining to the domains of emotion processing and social cognition, making the distribution of our social and emotion tasks along these gradients uninterpretable. Another possibility is that distribution of these two tasks along gradients 1 and 2 may indeed provide valid insights into the organization and nature of social and emotion processing in the cerebellum, as follows. Social processing task activity map spanned across Gradient 1, perhaps reflecting a multimodal nature of social processing in the cerebellum in the dimension of motor to nonmotor processing. The conceptualization of social processing in the cerebellum as an activity that engages multiple levels of information processing along the motor-nonmotor dimension may relate to the concomitant impairment of social skills, nonmotor tests such as Rey's figure or Tower test, and some motor abilities (e.g. equilibrium and limb coordination) in autism spectrum disorders (*Paquet et al., 2016*). Emotion processing was situated at a central position in both Gradient 1 and 2. We understand this distribution as an inability to clearly classify emotion processing along the gradients of distance from motor processing (Gradient 1) and amount of task focus (Gradient 2). Higher working memory load as isolated by the HCP working memory task corresponds to a level of information processing with high task-focus demands. At the same time, the subtraction of the HCP language task isolates task-unfocused processes which are maximally removed from pure motor processing. The results of the HCP emotion processing task contrast, on the other hand, are not as well defined along these dimensions. The subtraction of *Faces* ('decide which of two angry/fearful faces on the bottom of the screen match the face at the top of the screen') minus *Shapes* (same task performed with shapes instead of faces) isolates higher emotional content in the information that is processed. It may be argued that this higher emotional content corresponds to an intermediate position between pure motor and high-nonmotor level information processing (explaining the intermediate position along Gradient 1), and that this higher emotional content results in mildly increased task focus (explaining the intermediate position along Gradient 2).

## Confirmation of the double/triple representation hypothesis

Resting-state as well as task processing analyses have revealed a cerebellar double motor (lobules I-VI and VIII) and triple non-motor representation (lobules VI/Crus I, Crus II/VIIB and IX/X) (*Buckner et al., 2011*; *Guell et al., 2018a*). The distribution of Gradient 1, the component that explains the greatest variability in resting-state intra-cerebellar connectivity patterns, confirms this organization. Gradient 1 lowest values correspond to lobules IV/V/VI and VIII (*Figure 1A*, dark blue regions in Gradient 1), demarcating the two areas of motor representation. The highest values correspond to lobules Crus I, Crus II, and lobule IX (*Figure 1A*, dark red regions in Gradient 1) - these regions correspond to the first, contiguous second, and third nonmotor representation areas, respectively. Taken together, the double motor/triple nonmotor organization has now been shown in cerebellar representations of cerebral resting-state networks (*Buckner et al., 2011*), cerebellar task activity (*Guell et al., 2018a*), cerebro-cerebellar functional connectivity from cerebral cortical task activity peaks (*Guell et al., 2018a*), and gradients of intra-cerebellar patterns of functional connectivity (the present study). Gradient 2 also revealed a similar distribution, with its maximum values located in Crus I, Crus II/VIIB, and lobules IX/X.

A very similar organization was also found when calculating functional gradients based on connectivity from the cerebellum to the cerebral cortex (*Figure 2*). Clustering of connectivity gradients revealed discrete networks resembling cerebello-cerebral connectivity parcellations in *Buckner et al. (2011)*, and also replicated their double motor/triple nonmotor representation distribution (*Figure 1—figure supplement 5*). These observations support the generalizability of the double motor/triple nonmotor representation hypothesis to multiple directions of functional connectivity, namely, cerebello-cerebral and intra-cerebellar.

A 'network approach to the localization of complex functions' rather than 'an exclusive concentration of function within individual centers in the brain' (*Mesulam, 1981*) has long been adopted in the cerebral cortex (*Yeo et al., 2011*; *Mesulam, 1981*, *1986*; *Goldman-Rakic, 1988*), although some complex functions are indeed organized into focally specific brain regions (*Kanwisher et al., 1997*; *Saxe and Kanwisher, 2003*). Accumulating evidence for a double motor/triple nonmotor organization in the cerebellum warrants an analogous shift in the understanding of cerebellar functional neuroanatomy. Just as 'each distributed network consists of association areas spanning frontal, parietal, temporal and cingulate cortices' (*Mesulam, 1981*), the data indicate that each nonmotor cerebellar network consists of three representations spanning VI/Crus I, Crus II/VIIB and IX/X. There are no

intrinsic anatomical connections linking these cerebellar areas, but tract tracing studies in monkeys hint at the possibility of an anatomical correlate of the double motor/triple nonmotor organization. This conclusion is based on shared cerebello-cerebral cortical loops: lobules I-VI and VIII receive input from and project to M1, and lobules Crus I/Crus II and IX/X receive input from and project to area 46 (*Kelly and Strick, 2003*). Further, as in the cerebral cortex, distributed networks may exist adjacent to each other within each area of nonmotor representation in the cerebellum. In the same way that '*adjacent areas in the parietal cortex belonging to separate networks are differentially connected to adjacent areas of corresponding networks in the frontal, temporal and cingulate cortices*' (*Yeo et al., 2011*; *Selemon and Goldman-Rakic, 1988*; *Cavada and Goldman-Rakic, 1989a*, *1989b*), adjacent areas in VI/Crus I belonging to separate networks are differentially related to adjacent areas of corresponding networks in Crus II/VIIB and IX/X. This is revealed by non-overlapping nonmotor task activity maps within each area of representation in *Guell et al. (2018a)*, the unmasking of multiple resting-state networks within each area of representation in *Buckner et al. (2011)*, and the distribution of Gradient 1 in the present analysis. Task contrasts or connectivity analyses might reveal incomplete engagement of the triple nonmotor cerebellar network - a discussion regarding this incomplete engagement would be appropriate in these cases. For instance, incomplete engagement of the triple nonmotor network might be functionally meaningful, for example, activity in the areas of first and second representations, but not in the area of third representation. Similarly, future studies may discuss group contrasts where a given neurological or psychiatric disease results in functional or structural cerebellar abnormalities within only one area of motor or nonmotor representation. This approach might be critical for the understanding of cerebellar systems physiology and pathophysiology. Consequently, a critical next step towards a more comprehensive and nuanced understanding of cerebellar functional neuroanatomy is the investigation of distinct contributions of each area of motor and nonmotor representation. The following section addresses this question.

## Second motor (VIII) and third nonmotor representation regions (IX/X) are situated at a less extreme level along Gradient 1 and/or 2: Cerebellar motor and nonmotor representations are functionally distinct, and second motor representation might share functional similarities with third nonmotor representation

A review (*Sokolov et al., 2017*) frames the question of 'the functional significance of the two (or three) cortical representation maps in the cerebellum' as one of the principal outstanding enigmas in cerebellar neuroscience. Our present study provides the data to attempt to address this question for the first time, as follows.

Second motor representation (lobule VIII) and third nonmotor representations (lobule IX/X) were consistently located at less extreme positions along Gradient 1 and/or 2 when compared to their first motor and first/second nonmotor representations, respectively. This pattern was observed in all maps analyzed, including gradient-derived cerebellar parcellations (*Figure 3*), task activity maps (from *Guell et al., 2018a*, *Figure 3—figure supplement 1*), and resting-state maps (from *Buckner et al., 2011*, *Figure 3—figure supplement 1*). Further, this distribution was also observed in 15 min of resting-state data in a single subject (*Figure 3—figure supplement 4*). This observation indicates that the contribution of the second motor representation (lobule VIII) is different from the contribution of the first motor representation (lobules I-VI), as expected from previous clinical (*Stoodley et al., 2016*; *Schmahmann et al., 2009*) and functional connectivity (*Kipping et al., 2013*) observations. Crucially, it also indicates that the contribution of the third nonmotor representation (lobules IX/X) is different from the contribution of the first and second nonmotor representations (lobules VI/Crus I/Crus II/VIIB). These conservative conclusions are, on their own, novel in the field of cerebellar systems neuroscience. We further speculate that a less extreme position along Gradient 1 and/or 2 in both third nonmotor and second motor representation represents, in both cases, a less extreme level of information processing. 'Extreme' here refers to the poles of the sensorimotor-fugal organization (Gradient 1) and the task-focus/task-unfocus organization (Gradient 2). Specifically, a less extreme position along Gradient 1 corresponds to a less extreme level of information processing along the motor/nonmotor dimension, and a less extreme position along Gradient 2 corresponds to a less extreme level of information processing along the task-unfocused/task focused dimension. Because this pattern of a less extreme level of information processing is observed in both second

motor representation and third nonmotor representation, we argue that nonmotor activity in lobules IX and X (third nonmotor representation) might emerge from, and follow the logic of, motor processing in lobule VIII (second motor representation). This notion is inspired by the organization of the cerebral cortex where multimodal or association cortical areas are related to their nearby unimodal areas. For example, Broca's area is close to the primary motor cortex, while Wernicke's area is close to the primary auditory cortex. The analogy in the cerebellum is that nonmotor activity in lobules IX and X is adjacent to, and therefore follows the logic of, motor activity in lobule VIII. Restated, the relationship between first motor and second motor representation resembles the relationship between first/second nonmotor and third nonmotor representations, just as the relationship between primary motor and primary auditory cortex reflects the relationship between Broca's area and Wernicke's area.

The data show that the second representation of motor task activity, sensorimotor network, and 'Low-G1' (motor) maps were consistently located at a higher position along Gradient 2 when compared to their first representation. This suggests that while the first motor representation is engaged in pure motor processing as isolated by the *Movement* (e.g. tap left fingers) minus *Average* (average of the other four movements) contrast, second motor representation is engaged in motor processes that require higher task focus. In this way, second motor representation corresponds to a less extreme level of task-unfocused motor information processing. Following a similar logic, third representation of language task, DMN, and 'High-G1' maps were consistently located at a lower position along Gradient 1 when compared to their first and contiguous second representations. While these first and contiguous second representations are at an extreme level of information processing (i.e. maximally nonmotor), third representation is in a less extreme position (i.e. less extreme in the motor/nonmotor dimension). Also consistent with this logic, the third representation of working memory, frontoparietal network, and 'High-G2' maps were consistently located at a lower position along Gradient 2 when compared to their first and second representations. These first and second representations were at an extreme level of information processing – specifically, maximally task-focused. The third representation was located further from this extreme, that is, less extreme in the task-unfocused/task-focused dimension. Ventral and dorsal attention networks were not located at one clear gradient extreme, but their distribution of three representations also followed the logic that third representation (lobule IX/X) was located at a less extreme position along Gradient 1 and/or 2.

Of note, the second representation of working memory, frontoparietal network, and 'High-G2' was located similar to its first representation. This proximity between the first and second nonmotor representations indicates that the relationship between second motor and third nonmotor representation does not apply to the relationship between second motor and second nonmotor representation. Restated, nonmotor processes in lobules IX/X share hierarchical principles with motor processing in lobule VIII (an analogous 'less extreme' level of information processing) - in contrast, this relationship does not apply between nonmotor processing in lobules Crus II/VIIB and motor processing in lobule VIII.

A cerebello-cerebral connectivity analysis further supports the hypothesis that second motor and third nonmotor regions of representation correspond to a less extreme level of information processing when compared to their first motor and first/second nonmotor representations, respectively. These analyses are shown in *Figure 3—figure supplement 5*: while connectivity from first motor representation corresponds to cerebral cortical somatomotor network, connectivity from second motor representation engages areas adjacent to, but not directly at, somatomotor network (as shown previously by Kipping and colleagues [*Kipping et al., 2013*]). Similarly, while connectivity from first/second task-focused nonmotor representation corresponds to cerebral cortical task-positive networks, connectivity from third task-focused nonmotor representation engages areas adjacent to, but not directly at, task-positive networks. Following the same logic, while connectivity from first/second task-unfocused nonmotor representation corresponds to cerebral cortical task-negative networks (Default Mode Network), connectivity from third task-focused nonmotor representation engages areas adjacent to, but not directly at, task-negative networks.

The constellation of symptoms that follow cerebellar strokes of the posterior inferior cerebellar artery (PICA) may also support our hypothesis. PICA occlusion commonly results in the infarction of lobule VIII (second motor representation) but not of lobules IV/V/VI (first motor representation). Notably, these lesions result in little or no motor deficits (*Stoodley et al., 2016*;

*Schmahmann et al., 2009*). Our hypothesis that second motor representation corresponds to a less extreme level of pure motor information processing might explain the lack of pure cerebellar motor symptoms (gait ataxia, appendicular dysmetria, dysarthria) after PICA stroke. Whereas the pattern of deficits arising from lesions of the second motor representation may go undetected with the standard neurological motor examination, our data predict that fine discriminative testing may reveal deficits in motor-related tasks that require high task focus. This might include motor performance abnormalities that only manifest in the presence of distractors. However, PICA strokes also damage other lobules such as Crus II and VIIB - deficits in motor tasks requiring high task focus may be difficult to dissociate from nonmotor abnormalities arising from infarction of cerebellar regions other than lobule VIII. We are not aware of any report of isolated lobule VIII injury in humans - however, *Dow, (1938*) performed isolated ablation of lobule VIII in three rhesus monkeys. The author reported that '*In all three animals in which the pyramis* (i.e. lobule VIII) *alone was damaged little that was abnormal could be detected, except that the animal when running down a long corridor apparently was unable to stop quickly enough to avoid crashing head-on against the end wall. No visual defect was present. The abnormality was never observed later than the third or fourth day after operation*'. Aberrant motor behavior in the absence of classical cerebellar motor symptoms may be consistent with our reasoning. fMRI task activity analyses have made claims regarding distinct functional contributions of the cerebellar second motor representation (*Habas et al., 2004*; *Diedrichsen et al., 2005*; *Habas and Cabanis, 2006*; *Bohland and Guenther, 2006*; *Tourville et al., 2008*); however, none has demonstrated statistically significant lobule VIII activity in the absence of lobule IV/V/VI activity for any given task contrast. Kipping and colleagues (*Kipping et al., 2013*) reported lobule VIII functional connectivity with cerebral cortical regions other than motor and premotor regions, a pattern of connectivity consistent with our hypothesis that the second motor representation is located at a less extreme level of motor processing.

We showed that during a working memory task there was activity in the cerebellum in the first and second nonmotor representations, but not in the third representation (*Guell et al., 2018a*). In contrast, functional connectivity was observed in all three areas of representation when seeding from the cerebral cortical peak of the working memory task. In the light of the present observations, our interpretation is that functional connectivity revealed the full pattern of triple representation of task-focused mid-nonmotor processing areas, but when engaged with a working memory task, the third representation in the network was not recruited due to excessive task-focus demands (i.e. due to an extreme level of information processing along the task-unfocused/task-focused dimension).

Some anatomical peculiarities of lobules IX/X conform to the notion of a functionally distinct nonmotor contribution of these lobules. Glickstein and colleagues (*Glickstein et al., 1994*) reported that the principal target of pontine visual cells in monkeys is lobule IX. A specific type of cell, the Calretinin-positive unipolar brush cell, is preferentially located in lobules IX and X in many species (*Diño et al., 1999*; *Mugnaini et al., 2011*) and receives vestibular afferents (*Diño et al., 2001*). Accordingly, lobules IX and X are classically considered to represent the vestibulocerebellum. One highly speculative proposal is that the incorporation of visual/vestibular streams of information in lobules IX/X, but not in lobules VI/Crus I/Crus II/VIIB, might be related to the asymmetries we describe between the third and the first/second nonmotor representations. Indeed, some lines of study investigate the link between vestibular function and limbic and cognitive functions including visuospatial reasoning (*Hitier et al., 2014*; *Bigelow and Agrawal, 2015*; *Rajagopalan et al., 2017*). The notion that asymmetry between nonmotor representations may arise from heterogeneity in cerebellar patterns of connectivity, rather than cytoarchitecture or physiology, is in accord with the notion of a Universal Cerebellar Transform (*Schmahmann, 1996*, *Schmahmann, 1991*; *Guell et al., 2018b*).

## Concluding remarks and relevance for future investigations

This is the first study to describe the principal gradient of macroscale function in the cerebellum. Previous studies have segregated the cerebellum into discretely arranged functional regions (*Buckner et al., 2011*; *Guell et al., 2018a*). This leads to the fundamental question: What is the relationship between the cerebellar regions that subserve these distinct networks? Following a logic similar to the fundamental and well-established primary-unimodal-transmodal hierarchical organization in the cerebral cortex (*Mesulam, 1998*; *Margulies et al., 2016*), we report that cerebellar macroscale organization is sensorimotor-fugal. Regions further from the central aspect of lobules IV/V/VI

and VIII are, accordingly, further from cerebellar motor function in a gradient from motor to maximally non-motor (mind-wandering, non-goal oriented) function. This concept is analogous to the well-established knowledge in the cerebral cortex that regions progressively further from primary cortices (motor/somatosensory, auditory, visual) are progressively involved in more abstract, transmodal, non-primary processing. This fundamental concept has greatly influenced topographical investigations in the cerebral cortex, and it is reasonable to consider that the present description may equally influence cerebellar investigations. The publicly available cerebellum gradient maps from the present study in multiple file formats and structural spaces (https://github.com/xaviergp/cerebellum_gradients, folder 'FINAL_GRADIENTS') will facilitate the inclusion of the sensorimotor-fugal principle of cerebellar macroscale organization in future investigations.

The distribution of these principal gradients confirmed the double motor/triple nonmotor organization in the cerebellum, highlighting the need to refer to this organization when discussing cerebellar functional or structural findings. Close attention to this network organization may become critical for the understanding of cerebellar structure and function in health and disease. Clusters in lobules IV/V/VI and VIII are commonly interpreted as first and second representations of motor processing. The same reasoning should be applied to nonmotor findings, for example, in the interpretation of degeneration of Crus I and IX in Alzheimer's disease (Guo et al., 2016).

One important additional implication of the analysis of connectivity gradients in the present study is the unmasking of functional differences not only between the two motor cerebellar representations (as expected from previous clinical (Stoodley et al., 2016; Schmahmann et al., 2009) and functional connectivity (Kipping et al., 2013) observations), but also between the three nonmotor cerebellar representations (Figure 3, Figure 3—figure supplement 1). An initial hypothesis regarding the nature of these differences is generated by noting hierarchical similarities between second motor (VIII) and third nonmotor (IX/X) representations in gradient-derived parcellations, task activity, and resting-state maps. We interpret this relationship as an indication that nonmotor processing in lobules IX/X emerges from, and follows the logic of, motor processing in lobule VIII – specifically, processing in both regions corresponds to a less extreme level of information processing when compared to nonmotor processing in VI/Crus I/Crus II and motor processing in I-VI. This hypothesis may be useful in the interpretation of future cerebellar neuroimaging findings. For example, this hypothesis may help interpret or highlight the potential relevance of isolated abnormalities in lobule VIII and IX in ADHD (Hove et al., 2015). A virtue of this hypothesis is that it is testable using task fMRI. For example, future studies may contrast motor task conditions with high versus low task-focus demands (to isolate second motor representation), task-focused nonmotor task conditions with lower versus higher task-focus demands (to isolate third task-focused nonmotor representation), and task-unfocused nonmotor task conditions which can be removed from motor processing by, for example, modulating the amount of mental object manipulation (to isolate the third task-unfocused nonmotor representation).

In sum, we describe a fundamental sensorimotor-fugal principle of organization in the cerebellum, confirm the double motor/triple nonmotor representation organization, unmask functional differences not only between the two motor but also between the three nonmotor areas of representation, and hint at the possibility that second motor and third nonmotor representations might share functional similarities. Our findings and analyses represent a significant conceptual advance in cerebellar systems neuroscience, and introduce novel approaches and testable questions to the investigation of cerebellar topography and function.

## Materials and methods

All code used in this study is openly available at https://github.com/xaviergp/cerebellum_gradients (Guell, 2018; copy archived at https://github.com/elifesciences-publications/cerebellum_gradients).

### Human connectome project data

fMRI data were provided by the Human Connectome Project (HCP), WU-Minn Consortium (Van Essen et al., 2013). We analyzed data from 1003 participants who completed all resting-state sessions (age mean = 28.71, SD = 3.71, 470 male, 533 female), four 15 min scans per subject), included in the group average preprocessed dense connectome S1200 HCP release. Of note, many participants within this group of 1003 participants were related. There were 120 pairs of

monozygotic twins, and 64 pairs of dizygotic twins (as determined by genetic testing in the data provided by HCP). Of the 635 remaining individuals, there were 132 pairs of related participants, 37 groups of 3 related participants, 10 groups of 4 related participants, and 3 groups of 5 related participants. In total, there were 798 related and 205 unrelated participants. While brain organization is expected to be more similar between pairs of related subjects than between pairs of unrelated subjects, this feature of our data was not accounted for in our analysis. Analysis of a subset of 32 unrelated participants tested whether our findings were also observable in cohorts of unrelated subjects. EPI data acquired by the WU-Minn HCP used multi-band pulse sequences (*Moeller et al., 2010*; *Feinberg et al., 2010*; *Setsompop et al., 2012*; *Xu et al., 2013*). HCP structural scans are defaced using the algorithm by Milchenko and Marcus (*Milchenko and Marcus, 2013*). HCP MRI data preprocessing pipelines are primarily built using tools from FSL and FreeSurfer (*Glasser et al., 2013*; *Jenkinson et al., 2012*; *Fischl, 2012*). HCP structural pre-processing includes cortical myelin maps generated by the methods introduced by Glasser and Van Essen (*Glasser and Van Essen, 2011*). HCP task-fMRI analyses uses FMRIB's Expert Analysis Tool (*Jenkinson et al., 2012*; *Woolrich et al., 2001*). All group fMRI data used in the present study included 2 mm spatial smoothing and areal-feature aligned data alignment ('MSMAll') (*Robinson et al., 2014*). We did not conduct any further preprocessing beyond what was already implemented by the HCP. Results were visualized in volumetric space as provided by HCP as well as on a cerebellar flat map using the SUIT toolbox for SPM (*Diedrichsen, 2006*; *Diedrichsen et al., 2009*; *Diedrichsen and Zotow, 2015*).

## Diffusion map embedding

Diffusion map embedding methodology was introduced by Coifman and colleagues (*Coifman et al., 2005*), and its application to the HCP resting-state data as performed in this study is thoroughly described in *Margulies et al. (2016)*. Instead of analyzing data corresponding to the cerebral cortex (*Margulies et al., 2016*), the present study included only voxels corresponding to the cerebellum. We used data from the S1200 release (n = 1003) instead of the S900 release (n = 820). In brief, cerebellar data in the preprocessed HCP 'dense connectome' includes correlation values of each cerebellar voxel with the rest of cerebellar voxels. In this way, each cerebellar voxel has a spatial distribution of cerebellar correlations (a 'connectivity pattern'). Diffusion map embedding is a nonlinear dimensionality reduction technique and can be used to analyze similarities between functional connectivity based networks. As in Principal Component Analysis (PCA), diffusion map embedding results in a first component (or 'principal gradient') that accounts for as much of the variability in the data as possible. Each following component (each following gradient) accounts for the highest variability possible under the constraint that all gradients are orthogonal to each other. The final result of a dense connectome matrix PCA analysis would take the form of a mosaic; if this method was applied, each cerebellar voxel would be assigned to a particular network with discrete borders. In contrast, diffusion embedding extracts overlapping 'gradients' of connectivity patterns from the initial matrix. For example, in *Margulies et al. (2016)*, gradient 1 extended from primary cortices to DMN areas, gradient 2 extended from motor and auditory cortices to the visual cortex, etc. Each voxel is then assigned a position within each gradient. In *Margulies et al. (2016)*, a voxel corresponding to a DMN area would be assigned an extreme position in gradient 1 (e.g. a value of 6.7 in a unitless scale from −5.4 to 6.9) and a middle position in gradient 2 (e.g. a value of 1.8 in a unitless scale from −3.0 to 5.7). In this way, the result of diffusion embedding is not one single mosaic of discrete networks, but multiple, continuous maps (gradients). Each gradient reflects a given progression of connectivity patterns (e.g. from DMN to sensorimotor, from motor/auditory cortex to visual cortex, etc.), each gradient accounts for a given percentage of variability in the data, and each voxel has a position within each gradient. An illustration of this method is provided in *Figure 1—figure supplement 1*.

It is important to highlight that our initial dense connectome matrix includes the profile of connectivity of each cerebellar voxel with the rest of the cerebellum only, rather than with the rest of the brain. In this way, our analysis reflects the intrinsic organization of the cerebellum without invoking its connectivity profiles with the cerebral hemispheres or other brain structures. This approach allows the possibility of identifying cerebellar properties that might otherwise be obscured in whole-brain connectivity analyses. The latter approach would emphasize the relationship between cerebellar structures and cerebral resting-state networks, and potentially miss relevant gradients of connectivity patterns within cerebellar resting-state data.

Diffusion map embedding and task processing analyses were also performed using a 15 min resting-state run from a single subject. To avoid selection bias, we chose to analyze the HCP participant corresponding to the 'single subject' download package of the HCP database. Resting-state smoothing of single-subject data was performed on the resulting gradients after diffusion map embedding calculations to avoid introducing artefactual correlations. Group and single subject evaluation of an additional group of 32 participants was performed to explore the applicability of our method to smaller group investigations. These participants corresponded to the first 32 participants of the '100 unrelated' download package of the HCP database. Analysis of this subgroup of 32 participants was performed using a normalized and concatenated dataseries file of all 32 participants, including one 15 min resting-state run per subject. Gradients resulting from the 32-participants group analysis were not smoothed. Individual subject analyses within this subgroup of 32 participants were performed using one 15 min resting-state run per subject, and smoothing (sigma = 4) was performed on the resulting gradients after diffusion map embedding calculations.

There are no cerebellar cortical association fibers (*Schmahmann, 1996*; *Schmahmann and Pandya, 2008*). Intra-cerebellar functional gradients may be driven by cerebello-cerebral interactions, and it is therefore reasonable to consider that intra-cerebellar and cerebello-cortical functional gradients may be similar. For this reason, we also calculated functional gradients in the cerebellum using functional connectivity values from the cerebellum to the cerebral cortex (rather than within the cerebellum) in the average 1003 participants connectivity matrix.

## Task activity and resting-state network maps

Cerebellar task activity data from a subset of 787 HCP participants were analyzed in a previous study by our group (*Guell et al., 2018a*). Guell and colleagues (*Guell et al., 2018a*) provided Cohen's d task activity maps thresholded at 0.5 (medium effect size). A sample size of 787 participants ensures that a Cohen's d value higher than 0.5 will be statistically significant even after correction for multiple comparisons in the cerebellum. These tasks include the following contrasts: *Movement* (tap left fingers, or tap right fingers, or squeeze right toes, or squeeze left toes, or move tongue) minus *Average* (average of the other four movements), assessing motor function (*Buckner et al., 2011*); *Two back* (subject responds if current stimulus matches the item two back) minus *Zero back* (subject responds if current stimulus matches target cue presented at start of block), assessing working memory; *Story* (listen to stories) minus *Math* (answer arithmetic questions), assessing language processing (*Binder et al., 2011*); *TOM* (view socially interacting geometric objects) minus *Random* (view randomly moving geometric objects), assessing social cognition (*Castelli et al., 2000*; *Wheatley et al., 2007*); and *Faces* (decide which of two angry/fearful faces on the bottom of the screen match the face at the top of the screen) minus *Shapes* (same task performed with shapes instead of faces), assessing emotion processing (*Hariri et al., 2002*).

Cerebellar resting-state network maps were obtained from *Buckner et al. (2011)*. Buckner's study applied a winner-takes-all algorithm to determine the strongest functional correlation of each cerebellar voxel to one of the 7 or 17 cerebral cortical resting-state networks defined by Yeo and colleagues (*Yeo et al., 2011*). This analysis used data from 1000 participants. Additional methodological information regarding these task activity and resting-state network data is included in *Supplementary file 1*.

## Clustering analyses

Clustering analyses on the resulting diffusion map embedding gradients included k-means clustering, spectral clustering, and silhouette coefficient analysis (*Hastie et al., 2009*) using the scikit-learn toolbox (*Pedregosa et al., 2011*). K-means separates samples in a previously specified number of clusters, minimizes the sum of the squared differences of each data point from the mean within each cluster, but makes the assumption that clusters are convex. Spectral clustering does not have a convexity constraint, provides a valuable alternative method of analysis to validate k-means clustering results, but still requires a specification of a number of clusters. Silhouette coefficient analysis makes it possible to select the optimal number of clusters by optimizing the separation distance between clusters. We normalized the gradients prior to clustering when calculations included all 8 gradients; if normalization is not performed, gradient 1 obscures the contribution of the last gradients given its much larger range of values.

## Cerebello-cerebral functional connectivity

We aimed to compare asymmetries between the two motor (I-VI, VIII) and three nonmotor regions of cerebellar representation (VI/Crus I, Crus II/VIIB, IX/X) by comparing their relative position along diffusion embedding gradients. As a supplementary analysis, we also contrasted cerebello-cerebral connectivity from these regions using diffusion embedding gradient peaks within each of these areas of representation (e.g. contrasting cerebral cortical connectivity between first and second motor regions of representation). Cerebello-cerebral and intra-cerebellar connectivity Fisher's z transformed values were obtained from the preprocessed HCP 'dense connectome' (n = 1003); maps were contrasted using the method for comparing correlated correlation coefficients described by Meng and colleagues (*Meng et al., 1992*); and p maps were corrected for multiple comparisons within the cerebral cortex using $p < 0.05$ voxel-based false discovery rate calculations.

## Acknowledgements

This work was supported in part by the MINDlink foundation, La Caixa Banking Foundation (XG), and by NIH grants R01 EB020740 (SSG) and P41 EB019936 (SSG). Data were provided by the Human Connectome Project, WU-Minn Consortium (Principal Investigators: David Van Essen and Kamil Ugurbil; 1U54MH091657) funded by the 16 National Institutes of Health and Centers that support the Nation Institutes of Health Blueprint for Neuroscience Research; and by the McDonnell Center for Systems Neuroscience at Washington University.

## Additional information

### Funding

| Funder | Grant reference number | Author |
| --- | --- | --- |
| MINDlink Foundation | | Jeremy D Schmahmann |
| "la Caixa" Foundation | | Xavier Guell |
| National Institutes of Health | R01 EB020740 | Satrajit S Ghosh |
| National Institutes of Health | P41 EB019936 | Satrajit S Ghosh |

The funders had no role in study design, data collection and interpretation, or the decision to submit the work for publication.

### Author contributions

Xavier Guell, Conceptualization, Formal analysis, Writing—original draft; Jeremy D Schmahmann, John DE Gabrieli, Supervision, Writing—review and editing; Satrajit S Ghosh, Conceptualization, Software, Supervision, Writing—review and editing

### Author ORCIDs

Xavier Guell (iD) http://orcid.org/0000-0002-0684-0954
Satrajit S Ghosh (iD) http://orcid.org/0000-0002-5312-6729

### Ethics

Human subjects: Consent forms were previously approved by the Washington University Institutional Review Board as part of the Human Connectome Project.

### Decision letter and Author response

Decision letter https://doi.org/10.7554/eLife.36652.021
Author response https://doi.org/10.7554/eLife.36652.022

# Additional files

## Supplementary files

• Supplementary file 1. Methodological details of task activity and resting-state network maps.
DOI: https://doi.org/10.7554/eLife.36652.016

• Transparent reporting form
DOI: https://doi.org/10.7554/eLife.36652.017

## Data availability

All code used in the present study and main result files have been deposited in Github. fMRI data were provided by the Human Connectome Project.

The following previously published dataset was used:

| Author(s) | Year | Dataset title | Dataset URL | Database, license, and accessibility information |
|---|---|---|---|---|
| Van Essen DC, Ugurbil K | 2017 | Human Connectome Project | https://www.humancon-nectome.org/study/hcp-young-adult/document/1200-subjects-data-release | Publicly available at the cited URL (data use terms: https://www.humanconnectome.org/study/hcp-young-adult/data-use-terms) |

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
