## [Decision Letter]

Thank you for submitting your article "Functional gradients of the cerebellum" for consideration by *eLife*. Your article has been reviewed by three peer reviewers, and the evaluation has been overseen by Andreas Bostan (Guest Revieewing Editor) David Van Essen (Senior Editor).

The reviewers have discussed the reviews and the Reviewing Editor has drafted this decision letter to help you prepare a revised submission.

Summary:

The reviewers find that the manuscript submitted by Guell et al. describes novel and interesting findings on the functional organization of the cerebellum. They adapted a technique that was recently used by Marguiles et al. (2016) to describe a principal axis for spatial organization in the cerebral cortex (primary sensorimotor to multimodal regions). This mapping technique seems to be interesting and potentially useful to derive novel insights about the cerebellum. Based on their results, the authors put forth a novel framework for studying and understanding the functions of the cerebellar cortex. Three central conclusions seem essential for establishing this framework:

1) For a better understanding of cerebellar function, it is important to establish a functional parcellation of the cerebellar cortex, in contrast (and/or along) with partitioning it into discrete areas based on anatomical boundaries (e.g. cerebellar lobules).

2) Two main gradients for the organization of cerebellar function can be described. The first identifies a progression from uni-modal (motor) to trans-modal processing regions. The second identifies a progression from task-focused to task-unfocused regions.

3) The localization of the "double motor" and "triple non-motor" representations along the two main gradients indicate that the respective motor and non-motor sub-regions of the cerebellum are functionally distinct.

Overall, the reviewers would like the authors to have a more clear presentation of the results and a more focused discussion of their relevance.

Essential revisions:

1) The authors use several analyses to show that the organizational framework described here is very much in line with both task-fMRI activations and resting-state connectivity with the cerebral cortex. However, it would be useful for the authors to (more clearly and succinctly) emphasize the novel aspects of their work and the insights that their perspective adds to what we already know about the functional organization of the cerebellar cortex.

2) The methods discussing how the task-based and resting-state activations are integrated into this framework need to be clarified. Many methods are provided by reference only and not explained. Given the strong modeling focus of this manuscript, the reviewers suggest that the authors find a way to provide more details regarding their methodological choices.

3) One distinct aspect of their approach is the fact that they just focus on cerebellar resting-state activity, without invoking cerebellar connections with the rest of the brain. However, given the absence of intra-cerebellar connections, it is unclear how much of the functional connectivity within the cerebellum is driven by anything else but its connections with other regions of the brain, particularly the cerebral cortex. This raises the question of whether a full-brain analysis (rather than a cerebellum-specific analysis) would compare to the present results. The reviewers strongly suggest that the authors consider adding a cortical connectivity component to this manuscript.

4) An exciting aspect of the current manuscript is that the authors successfully replicate the results at the individual level. This suggests that this framework would be well suited for conducting small-sample studies of clinical populations. To emphasize this point, it would be nice to see more examples of replication at the individual level and/or replication in sub-samples. In order to encourage studies of clinical populations using this framework, it might also be useful if the authors provided an insight into the proportion of individuals in which the two main gradients can be reproduced at the individual level.

5) The authors discuss more than two gradients, but it is unclear how relevant their discussion is, given the limited variability that they explain and that they do not appear to replicate (well) at the individual level. Maybe this data could be included in just the supplementary information.

6) The notion of two main gradients of cerebellar organization indicates that some tasks (e.g. standard social/emotional processing tasks) may not be particularly well suited for identifying the functional role of the cerebellum. The authors may consider this (or a similar) perspective, instead of trying to fit the results from all functional tasks into their framework.

7) The results offer interesting insights regarding the double motor and triple non-motor representations. Although the nomenclature ("double" and "triple") suggests otherwise, the results indicate that the different cerebellar motor and non-motor representations are non-redundant and functionally distinct – this is a new and interesting idea that is worth both emphasizing and clarifying (Figure 2 presents these results, but is particularly difficult to follow with the given choices in color coding and nomenclature).

8) Several sections (particularly in the Introduction and Discussion) are fairly speculative and go beyond the scope of the manuscript. These distract the reader from the main points of the manuscript and should be simplified.

---

## [Author Response]

Essential revisions:1) The authors use several analyses to show that the organizational framework described here is very much in line with both task-fMRI activations and resting-state connectivity with the cerebral cortex. However, it would be useful for the authors to (more clearly and succinctly) emphasize the novel aspects of their work and the insights that their perspective adds to what we already know about the functional organization of the cerebellar cortex.

The Abstract and Discussion have been modified to better highlight the novel aspects of our work and its relationship with previous literature. New sentences include:

– New Abstract: “A central principle for understanding the cerebral cortex is that macroscale anatomy reflects a functional hierarchy from primary to transmodal processing. […] Functional differences exist not only between the two motor but also between the three nonmotor representations, and second motor representation might share functional similarities with third nonmotor representation.”

– Discussion: “Previous studies have segregated the cerebellum into discretely arranged functional regions (Buckner et al., 2011; Guell, Gabriel and Schmahmann, 2018). This leads to the fundamental question: What is the relationship between the cerebellar regions that subserve these distinct networks?”

– Discussion: “Further, the relationship between the two principal gradients and the two motor and three nonmotor areas of representation revealed for the first time that there are functional differences not only between the two motor but also between the three nonmotor areas of representation. An initial novel hypothesis regarding the nature of these differences is generated by noting that nonmotor processing in lobules IX/X (third nonmotor representation) might share functional similarities with motor processing in lobule VIII (second motor representation).”

– Discussion: “In sum, we describe a fundamental sensorimotor-fugal principle of organization in the cerebellum, confirm the double motor / triple nonmotor representation organization, unmask functional differences not only between the two motor but also between the three nonmotor areas of representation, and hint at the possibility that second motor and third nonmotor representations might share functional similarities.”

See also our response to comments 7 and 8, which also make reference to the need to emphasize the novelty of our work.

2) The methods discussing how the task-based and resting-state activations are integrated into this framework need to be clarified. Many methods are provided by reference only and not explained. Given the strong modeling focus of this manuscript, the reviewers suggest that the authors find a way to provide more details regarding their methodological choices.

Description of methods for task-based and resting-state maps that are used in Figure 1 has been expanded in a new supplementary table (Supplementary file 1).

3) One distinct aspect of their approach is the fact that they just focus on cerebellar resting-state activity, without invoking cerebellar connections with the rest of the brain. However, given the absence of intra-cerebellar connections, it is unclear how much of the functional connectivity within the cerebellum is driven by anything else but its connections with other regions of the brain, particularly the cerebral cortex. This raises the question of whether a full-brain analysis (rather than a cerebellum-specific analysis) would compare to the present results. The reviewers strongly suggest that the authors consider adding a cortical connectivity component to this manuscript.

To address this question we have calculated functional gradients in the cerebellum based on connectivity between the cerebellum and the cerebral cortex (rather than within the cerebellum). Our results indicate that functional gradients of cerebello-cerebral cortical connectivity are very similar to functional gradients of intra-cerebellar connectivity. New jupyter notebooks corresponding to these additional analyses have been added to the github repository. This finding has now been included as a figure in the main text of the manuscript (Figure 2).

New text in Materials and methods section: “There are no cerebellar cortical association fibers (Schmahmann, 1996; Schmahmann and Pandya, 2008). […]For this reason, we also calculated functional gradients in the cerebellum using functional connectivity values from the cerebellum to the cerebral cortex (rather than within the cerebellum) in the average 1003 participants connectivity matrix.”

New text in the Discussion: “Unlike the cerebral cortex, there are no short or long cerebellar cortical association fibers linking adjacent or distant cerebellar cortical areas with each other (Schmahmann, 1996; Schmahmann and Pandya, 2008). […] Consistent with this possibility, cerebellar functional gradients calculated from cerebello-cerebral connectivity revealed a distribution similar to the intra-cerebellar functional gradients (Figure 2).”

4) An exciting aspect of the current manuscript is that the authors successfully replicate the results at the individual level. This suggests that this framework would be well suited for conducting small-sample studies of clinical populations. To emphasize this point, it would be nice to see more examples of replication at the individual level and/or replication in sub-samples. In order to encourage studies of clinical populations using this framework, it might also be useful if the authors provided an insight into the proportion of individuals in which the two main gradients can be reproduced at the individual level.

Materials and methods section, text added: “Group and single subject evaluation of an additional group of 32 participants was performed to explore the applicability of our method to smaller group investigations. […] Individual subject analyses within this subgroup of 32 participants was performed using one 15-minutes resting-state run per subject, and smoothing (σ=4) was performed on the resulting gradients after diffusion map embedding calculations.”

Results section, text added: “Functional gradients calculated using concatenated and normalized time series of 32 participants also revealed a similar distribution of gradients 1 and 2 (Figure 1—figure supplement 4). […] Future studies aiming to perform group comparison statistics using functional gradients might benefit from alternate alignment strategies (see details in legend of Figure 1—figure supplement 4).”

The fact that our results remain observable at the individual subject level has been removed from the Abstract and first paragraph of the Discussion, and discussed only in the sections of the manuscript that make specific reference to these analyses.

Results and discussion are presented in a new supplementary figure (Figure 1—figure supplement 4).

5) The authors discuss more than two gradients, but it is unclear how relevant their discussion is, given the limited variability that they explain and that they do not appear to replicate (well) at the individual level. Maybe this data could be included in just the supplementary information.

Figures and result text for gradients other than gradients 1 and 2 have been removed from the main text, and a shortened version of this discussion is now only mentioned in the legend of supplementary figures.

Figure 1 now shows only gradients 1 and 2.

Additional gradients have been moved to a supplementary figure, Figure 1—figure supplement 2.

6) The notion of two main gradients of cerebellar organization indicates that some tasks (e.g. standard social/emotional processing tasks) may not be particularly well suited for identifying the functional role of the cerebellum. The authors may consider this (or a similar) perspective, instead of trying to fit the results from all functional tasks into their framework.

We have acknowledged the possibility that distribution of these two tasks might be uninterpretable, and added that as an alternative explanation to our initial discussion:

“While HCP motor, working memory and language task activity maps were situated at extreme regions along Gradient 1 and/or 2 (Figure 1B), social and emotion processing did not adhere to any extreme along these gradients. […] It may be argued that this higher emotional content corresponds to an intermediate position between pure motor and high-nonmotor level information processing (explaining the intermediate position along Gradient 1), and that this higher emotional content results in mildly increased task focus (explaining the intermediate position along Gradient 2).”

7) The results offer interesting insights regarding the double motor and triple non-motor representations. Although the nomenclature ("double" and "triple") suggests otherwise, the results indicate that the different cerebellar motor and non-motor representations are non-redundant and functionally distinct – this is a new and interesting idea that is worth both emphasizing and clarifying (Figure 2 presents these results, but is particularly difficult to follow with the given choices in color coding and nomenclature).

The main message of Figure 2 was presented in panels 2A and 2B. Given that panel C of the old version of Figure 2 was not central to the principal message of the figure, panel C of Figure 2 has been moved to a supplementary figure (Figure 3—figure supplement 1). This decreases the visual load in Figure 2, improving clarity. Panel B has now been subdivided into panels B and C, and text within Figure 2 has been changed to clarify the steps that are followed in the analysis presented in Figure 2. Legend of Figure 2 has also been clarified, and the notion that the different cerebellar motor and non-motor representations are non-redundant and functionally distinct has been emphasized in the Abstract, figure legend, and the first and last sections of the Discussion.

The fact that different cerebellar motor and non-motor representations are non-redundant and functionally distinct is emphasized in the Materials and methods: “This observation indicates that the contribution of the second motor representation (lobule VIII) is different from the contribution of the first motor representation (lobules IV/V/VI), as expected from previous clinical (Stoodley et al., 2016; Schmahmann et al., 2009)and functional connectivity (Kipping et al., 2013) observations. […] We further speculate that a less extreme position along Gradient 1 and/or 2 in both third nonmotor and second motor representation represents, in both cases, a less extreme level of information processing. “Extreme” here refers to …”

We have changed the title of the Discussion section to reflect the centrality of this concept:“Second motor (VIII) and third nonmotor representation regions (IX/X) are situated at a less extreme level along Gradient 1 and/or 2: Cerebellar motor and nonmotor representations are functionally distinct, and second motor representation might share functional similarities with third nonmotor representation.”

This concept has also been included in the first paragraph of the Discussion, as well as in the last section of the Discussion: “One important secondary implication of the analysis of connectivity gradients in the present study is the unmasking of functional differences not only between the two motor cerebellar representations (as expected from clinical observations (Stoodley et al., 2016; Schmahmann et al., 2009)), but also between the three nonmotor cerebellar representations (Figure 3, Figure 3—figure supplement 1).”

This concept has also been highlighted in the Abstract: “Further, these two principal gradients revealed novel functional properties of the well-established cerebellar double motor representation (lobules I-V and VIII), and its relationship with the recently described triple nonmotor representation (lobules VI-Crus I, Crus II/VIIB, IX/X). Functional differences exist not only between the two motor but also between the three nonmotor representations, and second motor representation might share functional similarities with third nonmotor representation.”

8) Several sections (particularly in the Introduction and Discussion) are fairly speculative and go beyond the scope of the manuscript. These distract the reader from the main points of the manuscript and should be simplified.

See our response to comment number 1 and 7, which also make reference to the way our results are discussed.

We agree that our discussion covers some aspects that are not indispensable for the main messages of the paper. Examples of these include paragraphs that discuss the following aspects:

1) The general distribution of datapoints in gradients 1 and 2 (Discussion, paragraph starting with “Our interpretation of task focus in the cerebellum in terms of distance from motor processing and amount of task-focus is also coherent with the general distribution of data points when plotting…”)

2) Network approaches in the cerebral cortex and its parallelisms with the cerebellum (Discussion, paragraph starting with “A network approach to the localization of complex functions…”)

3) Neurological symptomatology in cases of posterior inferior cerebellar artery stroke and findings in monkey lesion studies by Dr. Dow (Discussion, paragraph starting with “The constellation of symptoms that follow cerebellar strokes of the posterior inferior cerebellar artery…”)

4) Specific map of working memory task in Guell et al., 2018 (Discussion, paragraph starting with “We showed that during a working memory task there was activity…”)

5) Anatomical peculiarities of lobules IX/X (Discussion, paragraph starting with “Some anatomical peculiarities of lobules IX/X conform to the notion of a functionally distinct nonmotor contribution of these lobules”)

We nevertheless hope that the reviewers will agree that these paragraphs contribute to the quality of the manuscript, providing a detailed analysis that may be of interest to readers. Our approach has been to expand and reinforce the text that covers the relevance of our study, its relationship with previous literature, and relevance for future investigations. At the same time, we have not eliminated these less central aspects of our discussion, which we believe contribute to the quality of the manuscript.